# ISAC: Training-Free Instance-to-Semantic Attention Control for Improving Multi-Instance Generation

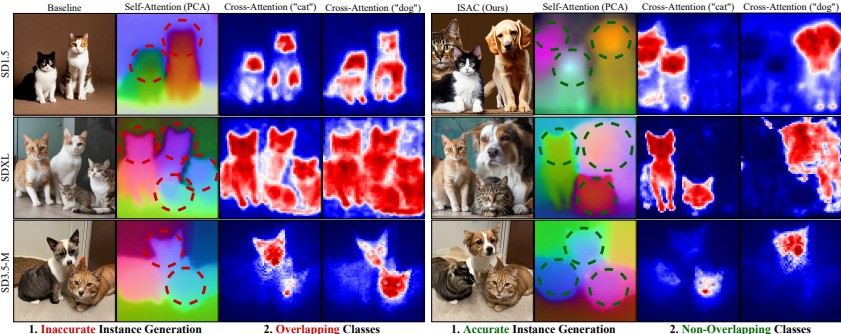

Figure 1: Comparison between existing text-to-image diffusion models (Rombach et al., 2022; Podell et al., 2023; Esser et al., 2024) (left) and the proposed ISAC framework (right) on the prompt "A photo of two cats and a dog". Text-to-image diffusion models struggle to clearly separate individual instances, leading to merged or overlapped objects (red dashed circles). In contrast, ISAC explicitly utilizes early-stage self-attention to accurately identify distinct, non-overlapping instances (green dashed circles) and assigns them precise semantic labels, resulting in clear class boundaries without additional training or supervision.

## Abstract

Text-to-image diffusion models excel at synthesizing single objects but frequently fail in multi-instance scenes, producing merged or missing objects. We show that this limitation arises because instance structures emerge before semantic features during denoising, making early semantic guidance unreliable. To address this, we propose **I**nstance-to-**S**emantic **A**ttention **C**ontrol (ISAC), a training-free and hierarchical inference objective that first enforces non-overlapping instance formation with self-attention and then aligns semantics through cross-attention. ISAC introduces a maximum pixel-wise overlap (MPO) criterion to strictly decouple instances and can be applied either as latent optimization or latent selection. Experiments on T2I-CompBench, HRS-Bench, and a new similar-object benchmark show that ISAC substantially improves both multi-class and multi-instance fidelity, achieving up to 52% multi-class accuracy and 83% multi-instance accuracy without external supervision. Our findings highlight the importance of aligning control with diffusion dynamics for faithful and scalable multi-object generation. The code will be made available upon publication.

## 1 Introduction

Text-to-image (T2I) diffusion models (Rombach et al., 2022; Podell et al., 2023; Esser et al., 2024; Chen et al., 2023a; Labs, 2024) have demonstrated remarkable capabilities in generating high-quality images from textual descriptions. While these models excel at generating single objects, they often fail in multi-instance scenes, omitting instances or merging them (e.g., "A photo of two cats and a dog"; Figure 1).

Prior work attributes this issue to the failure of diffusion models to assign distinct spatial regions to individual instances. To address this, recent methods have attempted to enforce separation by utilizing semantic signals—either by manipulating cross-attention maps within the UNet (Chefer et al., 2023; Rassin et al., 2024; Guo et al., 2024; Hu et al., 2024; Meral et al., 2024; Qiu et al., 2025; Wang et al., 2024b; Jiang et al., 2024) or by adjusting per-instance text embedding (Feng et al., 2023; Hu et al., 2024; Chen et al., 2024a). However, these approaches are fundamentally limited because the semantic signals they rely on are unreliable and not yet well-formed early in denoising. Consequently, relying on these underdeveloped signals during this critical period, when the image's core structure is established, often leads to the aforementioned failures.

In this paper, we discover that distinct semantics emerge *after* spatial instance structure forms, with diffusion dynamics analysis. Guided by this, we introduce **I**nstance-to-**S**emantic **A**ttention **C**ontrol (ISAC), a *hierarchical, two-phase* objective: (1) form $N$ non-overlapping instance structures from object counts; (2) bind semantics to those structures. Furthermore, to enable stricter separation than standard Intersection-over-Union (IoU), we propose a *maximum pixel-wise overlap (MPO)* criterion. We demonstrate the utility of the ISAC objective in two practical algorithms: latent optimization (using ISAC as a loss) and latent selection (using ISAC as a verifier).

We evaluate ISAC on the widely adopted T2I-CompBench (Huang et al., 2025), HRS-Bench (Bakr et al., 2023) benchmarks and an additional benchmark focused on similar-object scenarios. Across these settings, ISAC substantially improves multi-instance fidelity relative to recent approaches that do not account for diffusion dynamics. Because ISAC is model-agnostic, it also augments layout-guided pipelines. Qualitative results confirm these findings that ISAC remains highly beneficial even when instance locations are predefined.

To summarize, the key contributions of this work are:

- A novel analysis of the diffusion process for multi-instance generation that reveals a key temporal dynamic: distinct *instance structures* form early in the denoising process, well before cohesive *semantic features* emerge.
- Instance-to-Semantic Attention Control (ISAC), a novel *hierarchical, two-phase objective* designed around this dynamic, which prioritizes structural separation before enforcing semantic alignment.
- A new separation criterion, *maximum pixel-wise overlap (MPO)*, designed to enforce stricter instance boundaries than standard metrics like Intersection-over-Union (IoU).
- ISAC, as a *model-agnostic add-on*, consistently outperforms state-of-the-art methods on standard benchmarks and enhances existing layout-guided models.

## 2 RELATED WORK

A central challenge in text-to-image diffusion is the assignment of mutually exclusive spatial regions to multiple instances. Recent approaches seek to improve a model's regional awareness and can be grouped into three categories.

**Training-free Methods.** Cross-attention has been established as the primary interface between textual semantics and spatial layout (Hertz et al., 2022). Building on this insight, several works manipulate cross-attention maps to encourage non-overlapping instance regions. Attend-and-Excite (Chefer et al., 2023) maximizes attention peaks for neglected objects, and InitNO (Guo et al., 2024) additionally incorporates self-attention maps to consider spatial features. On the other hand, SynGen (Rassin et al., 2024) and CONFORM (Meral et al., 2024) introduced a contrastive loss that aims to separate distinct instances while binding visual attributes to their corresponding instances, aided by a syntax parser (Honnibal, 2017). Self-Cross (Qiu et al., 2025) extends these works by additionally incorporating self-attention maps into contrastive guidance. A complementary line of works adjusts per-instance text embeddings or token ordering to influence attention behavior and thereby improve spatial separation (Feng et al., 2023; Hu et al., 2024; Chen et al., 2024a). Another approach introduces guidance from a pretrained vision model rather than relying solely on internal representations (Kang et al., 2025).

Despite these advances, training-free methods generally depend on semantic signals that are unreliable in the early denoising steps, precisely when global structure is being established. Since utilizing incorrect cross-attention maps during structure formation can lead to failure—and applying guidance

from pretrained models is ineffective after these structures are already formed—a new paradigm for instance control that considers diffusion dynamics is needed.

**Fine-tuning Methods.** Several approaches incorporate explicit segmentation signals through model adaptation. Methods such as TokenCompose (Wang et al., 2024b) and CoMat (Jiang et al., 2024) fine-tune the UNet so that its cross-attention maps align with segmentation masks obtained from a pretrained model (Ren et al., 2024). Alternatively, CountGen (Binyamin et al., 2024) fine-tunes the diffusion model to first generate segmentation masks and then uses them to guide the final image synthesis. However, a significant drawback of these techniques is their limited training vocabulary compared to the original models, which restricts their general applicability. The proposed ISAC objective is complementary to such methods and can be employed at inference time with or without fine-tuned components.

**Layout-to-Image Methods.** Recent systems adopt a two-stage pipeline in which a layout of bounding boxes is generated from a text prompt and then used to guide image synthesis (Lian et al., 2023; Zhang et al., 2024a). State-of-the-art controllers provide dense layout control (Li et al., 2023; Zhou et al., 2024a; Wang et al., 2024a; Cheng et al., 2024; Zhou et al., 2024b), yet they frequently struggle when objects are adjacent because explicit mechanisms for instance separation are absent. As a model-agnostic addition, ISAC directly enforces separation among neighboring instances and thereby improves the reliability of layout-conditioned generation.

## 3 PRELIMINARIES

We address the text-to-image (T2I) generation task using latent diffusion models. Given a text prompt $C$, which we assume provides a set of class tokens $\{\tau_i\}_{i=1}^k$ and their corresponding instance counts $\{n_i\}_{i=1}^k$, the model generates a corresponding image.

### 3.1 LATENT DIFFUSION MODELS

Latent Diffusion Models (LDMs) learn to reverse a forward noising process that gradually corrupts a clean image latent $X_0$ into random noise $X_T$. A neural network $\epsilon_\theta$ is trained to predict the noise added at any timestep $t$, conditioned on the noisy latent $X_t$ and a text embedding $\mathcal{T} \in \mathbb{R}^{L \times d}$. The model is optimized with the following objective: $\theta^\star = \underset{\theta \in \Theta}{\operatorname{argmin}} \, \mathbb{E}_{(X_0, \mathcal{T}), \epsilon, t} \left[ \|\epsilon_\theta(X_t, t, \mathcal{T}) - \epsilon\|_2^2 \right]$.

**Sample Generation.** During inference, an image is synthesized with iterative denoising steps, $X_{t-1} \leftarrow \text{Denoise}(X_t, \mathcal{T}, \epsilon_\theta, t)$, starting from random noise $X_T \sim \mathcal{N}(0, I)$ to recover a clean latent $\hat{X}_0$. A VAE decoder $\mathcal{D}$ maps latent space to pixel space, obtaining a final image ($\hat{I} = \mathcal{D}(\hat{X}_0)$). Our framework is agnostic to generative dynamics (e.g., DDPM (Ho et al., 2020), Flow Matching (Esser et al., 2024)) and only requires access to the latent $X_t$ and the model's attention layers at each step.

**Architecture and Attention.** The denoiser $\epsilon_\theta$ is typically a U-Net or Diffusion Transformer (DiT) containing multiple attention layers. The core of the denoiser uses two key attention mechanisms: *self-attention*, which captures spatial relationships within the image latent, and *cross-attention*, which aligns spatial features with the text embeddings.

Let $X_t \in \mathbb{R}^{HW \times d}$ be the reshaped latent and $\mathcal{T} \in \mathbb{R}^{L \times d}$ be the text embedding. For a given attention head, query ($Q_t$) and key ($K_t$) vectors are computed using learned projection matrices. For *self-attention*, both are derived from the latent: $Q_t^{\text{self}} = X_t W_Q^{\text{self}}$, $K_t^{\text{self}} = X_t W_K^{\text{self}}$, and then computed as $SA(X_t) = \text{softmax}\big(Q_t^{\text{self}} K_t^{\text{self}\top} / \sqrt{d_h}\big) \in [0, 1]^{HW \times HW}$. For *cross-attention*, the query comes from the latent and the key from the text: $Q_t^{\text{cross}} = X_t W_Q^{\text{cross}}$, $K_t^{\text{cross}} = \mathcal{T} W_K^{\text{cross}}$, and then computed as $CA(X_t, \mathcal{T}) = \text{softmax}\big(Q_t^{\text{cross}} K_t^{\text{cross}\top} / \sqrt{d_h}\big) \in [0, 1]^{HW \times L}$.

### 3.2 ATTENTION ACCUMULATION VIA HOOKS

During sampling we register forward hooks, $\mathcal{H}^{\text{self}}$ and $\mathcal{H}^{\text{cross}}$, on all attention layers to extract attention maps without altering the computation. Let there be $M$ attention layers and $h_l$ attention heads at the $l$-th layer. Since the denoiser contains attention maps with various spatial resolutions, we upsample the maps from each layer $l$ and head $h$ to the highest spatial resolution $H \times W$. These maps are then averaged to produce a single accumulated attention map for each type:

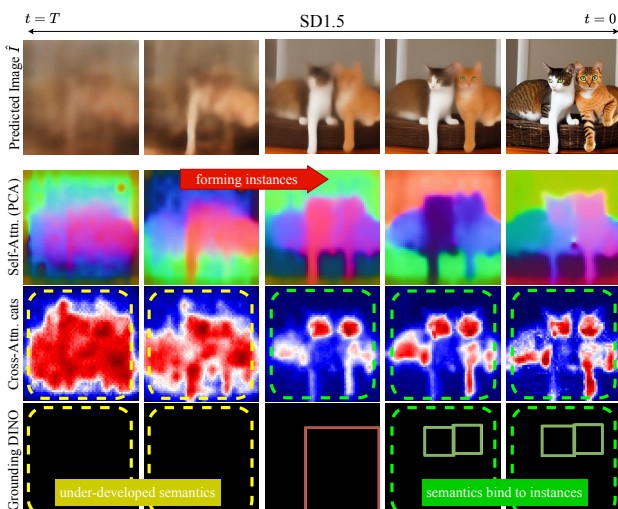

Figure 2: Temporal dynamics of diffusion models; Instance forms first, then semantic binds to it.

$$\mathcal{H}^{\text{self}}(X_t, \mathcal{T}, \epsilon_\theta, t) = \frac{1}{\sum_{l=1}^M h_l} \sum_{l=1}^M \sum_{h=1}^{h_l} \texttt{Upsample}(SA_l^h, \delta_l) \in \mathbb{R}^{HW \times HW} \tag{1}$$

$$\mathcal{H}^{\text{cross}}(X_t, \mathcal{T}, \epsilon_\theta, t) = \frac{1}{\sum_{l=1}^M h_l} \sum_{l=1}^M \sum_{h=1}^{h_l} \texttt{Upsample}(CA_l^h, \delta_l) \in \mathbb{R}^{HW \times L} \tag{2}$$

where $\delta_l = H/H_l = W/W_l$ is the upsampling factor for the $l$-th layer.

# 4 DISCOVERING DYNAMICS OF DIFFUSION MODELS IN MULTI-INSTANCE GENERATION

Figure 2 describes temporal dynamics of diffusion models. Before instance structures are formed, semantics of each instance is under-developed. Therefore, its internal representations, cross-attention maps, easily flood into the whole image. Also in this step, detection model (Liu et al., 2024) cannot find any instances because no recognizable semantic signals arise. When instance structures are formed and stabilized, their corresponding semantic signals bind to the instance and can also be recognizable by detection model. In Section E we show this is a model-agnostic behavior in the diffusion models family.

# 5 ISAC: DYNAMICS ALIGNED INSTANCE CONTROL OBJECTIVE

We propose a two-phase objective that first forms instance structures then binds semantics; it serves as a loss or a verifier. Figure 3 shows the overview of this objective.

## 5.1 PHASE 1: FORMING INSTANCE STRUCTURES WITH ONLY OBJECT COUNTS

When instance structures begin to form, semantic signals such as cross-attention maps from corresponding instances are still underdeveloped. Therefore, in Phase 1, instance separation must be performed using only structural signals. Suppose we target $k$ classes, represented by class tokens $\tau_1, \ldots, \tau_k$ and a total of $N = n_1 + \cdots + n_k$ instances. With only structural signals early on, we can conclude that "A total of $N$ instances should occupy mutually exclusive, $N$ distinct regions.".

To formulate this objective, we first create a *global foreground mask* to isolate relevant image regions. We then apply K-means clustering to the self-attention features within this foreground mask.

**Details on obtaining a global foreground mask** $M_{\text{fg}}$**.** Let $CA_t \in [0, 1]^{HW \times L}, SA_t \in [0, 1]^{HW \times HW}$ denote the accumulated cross- and self-attention map at timestep $t$, respectively, where $L$ is the length of the prompt token sequence. Following (Binyamin et al., 2024), we obtain a

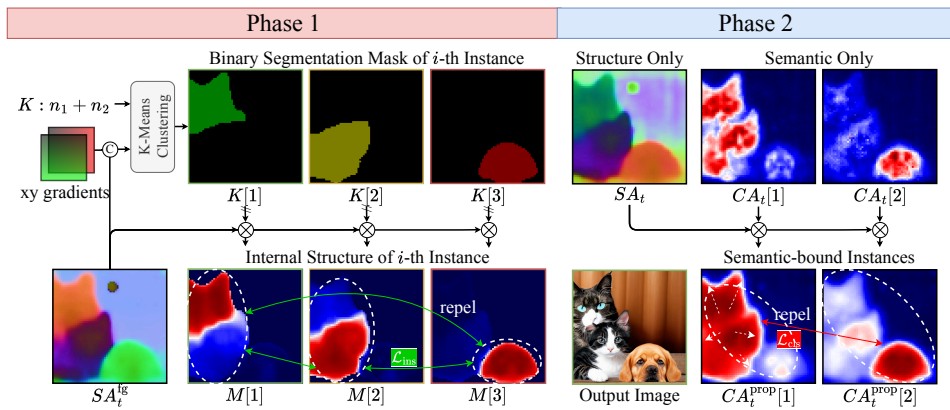

Figure 3: Overview of two phase control objective, ISAC. At phase 1, only the object counts work as a signal to separate instances, forming an objective $\mathcal{L}_{ins}$. At phase 2, where instance structures are formed, semantic signals are bind to instance structures. Now the phase 2 objective $\mathcal{L}_{cls}$ ensures each object semantic corresponds to the most probable instances.

*global foreground mask* by self-to-cross class propagation [1] (Equation 3) followed by column-wise adaptive binarization (Equation 4):

$$CA_t^{\text{prop}}(X_t, \mathcal{T}) \leftarrow SA_t \cdot CA_t \in [0,1]^{HW \times L} \tag{3}$$

$$CA_t^{\text{bin}}(X_t, \mathcal{T}) \leftarrow \texttt{Binarize}(CA_t^{\text{prop}}),$$
$$\text{where } \texttt{Binarize}(CA_t^{\text{prop}})[i,j] = \mathbf{1}[CA_t^{\text{prop}}[i,j] > \mu_j] \in \{0,1\} \tag{4}$$

where $\mu_j$ is the column-wise mean. The binarized map $CA_t^{\text{bin}} \in \{0,1\}^{HW \times L}$ contains foreground masks for target tokens $\tau_1, \ldots, \tau_k$ as well as non-target tokens (e.g., "the", "and"). The *global foreground mask* is the union of target tokens' masks: $M_{\text{fg}} = \bigcup_{\mathcal{T}[i] \in \{\tau_j\}_{j=1}^k} CA_t^{\text{bin}}[:, i] \in \{0,1\}^{HW}$.

**Achieving instance structures with K-means clustering.** From the global foreground mask $M_{\text{fg}}$, we index the self-attention to obtain the *filtered self-attention map* $SA_t^{\text{fg}}$ as in equation 6:

$$\mathcal{I} = \{i : M_{\text{fg}}[i] = 1\}, \quad SA_t \leftarrow \mathcal{H}^{\text{self}}(X_t, \epsilon_\theta, t) \tag{5}$$

$$SA_t^{\text{fg}} \leftarrow SA_t[\mathcal{I}, \mathcal{I}] \in [0,1]^{F \times F}, \text{ where } F := |\mathcal{I}| \tag{6}$$

For robust clustering, we concatenate to each row of $SA_t^{\text{fg}}$ the corresponding normalized image coordinates $(x, y) \in [-1, 1]^2$ (one scalar per axis). We then apply K-means with $K = N$ to these augmented vectors, producing a one-hot assignment matrix $K \in \{0, 1\}^{F \times N}$ (Figure 4).

This yields cluster-wise structures via a single matrix product: $SA_t^{\text{fg}} K \in [0, 1]^{F \times N}$. The $i$-th column of $SA_t^{\text{fg}} K$ aggregates dependencies of pixels assigned to cluster $i$, and we treat it as a soft instance mask $M[i]$, can be interpreted as the internal structure of the $i$-th instance.

**Measuring separation with Maximum Pixel-wise Overlap (MPO).** Let $A, B \in [0, 1]^F$ be soft masks over $F$ pixels. We define

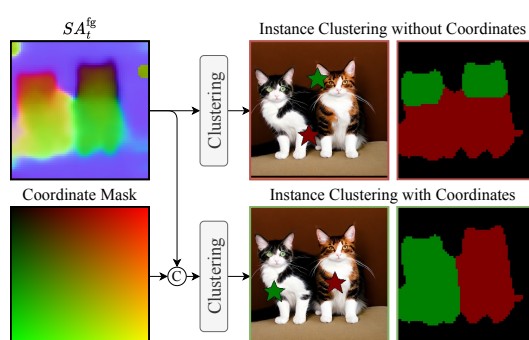

Figure 4: Adding X/Y coordinate masks improves instance clustering by introducing spatial cues, avoiding incorrect merging of separate instances.

$$\texttt{MPO}(A, B) = \max_{p \in \{1, \ldots, F\}} (A[p] \cdot B[p]) \tag{7}$$

---

[1] Class propagation is commonly used in segmentation methods (Shen et al., 2024; Wang et al., 2025b; Kipf & Welling, 2016; Zhu & Koniusz, 2021). We follow Shen et al. (2024)'s implementation.

which captures the peak local co-activation (worst-case overlap). Minimizing MPO suppresses even small but sharp collisions between masks, promoting spatial exclusivity that global similarities (e.g., IoU, KL) may under-penalize. We adopt MPO for both instance separation and semantic binding via $\mathcal{L}_{\text{ins}}$ and $\mathcal{L}_{\text{cls}}$; ablations and metric comparisons appear in §6.3 and Figure 6.

The Phase 1 objective, $\mathcal{L}_{\text{ins}}$, is therefore defined as the maximum MPO between any pair of instance structures:

$$\mathcal{L}_{\text{ins}}(X_t) = \max_{1 \leq i < j \leq N} \text{MPO}(M[i], M[j]) \tag{8}$$

## 5.2 Phase 2: Binding Semantics to Instance Structures

Once instance structures stabilize, we bind class semantics to the formed regions following the rationale: "the semantic appearance of each instance should be bound to its corresponding instance structure." We obtain semantic-bound maps by masking cross-attention with the formed structures (Equation 3), yielding $CA_t^{\text{cls}} \in [0,1]^{HW \times k}$ for classes $\tau_1, \ldots, \tau_k$. Let $\text{sign}(\tau_i, \tau_j) \in \{+1, -1\}$ encode the desired interaction between tokens: $+1$ for pairs that should *not* co-refer (different classes), $-1$ for pairs that *should* co-refer (attribute–object). Our Phase-2 objective is

$$\mathcal{L}_{\text{cls}}(X_t) = \max_{1 \leq i < j \leq k} \text{sign}(\tau_i, \tau_j) \cdot \text{MPO}(CA_t^{\text{cls}}[i], CA_t^{\text{cls}}[j]), \tag{9}$$

which penalizes peak overlap across different classes while encouraging peak co-activation for intended attribute binding. The relationship between tokens—$\text{sign}(\tau_i, \tau_j) \ \forall i, j$—can be easily achieved with modern large language models (LLMs) as a syntax parser.

## 5.3 Algorithmic Applications

The overall two-phase control objective can be written in a generalized form:

$$\mathcal{L}_t(X_t) := \lambda_{\text{ins}}(t)\mathcal{L}_{\text{ins}}(X_t) + \lambda_{\text{cls}}(t)\mathcal{L}_{\text{cls}}(X_t) \tag{10}$$

If we denote the phase transition timestep as $t^\star$, marking when instance formation is complete, the objective scheduling for this *hard* transition is defined with Heaviside step function $H$:

$$\lambda_{\text{ins}}(t) = 1 - H(t - t^\star), \quad \lambda_{\text{cls}}(t) = H(t - t^\star) \tag{11}$$

As the choice of schedule is flexible, we also propose a *soft* phase transition schedule, such as $\lambda_{\text{inst}}(t) = t/T, \lambda_{\text{cls}}(t) = 1 - t/T$, which eliminates the need to define $t^\star$ in advance. In our experiments, we empirically find that this *soft* schedule consistently outperforms the *hard* schedule from equation 11 when using $t^\star = T/2$.

The ISAC objective can be directly injected to guide the generation of multiple instances in two ways. First, similar to previous methods (Chefer et al., 2023; Guo et al., 2024; Meral et al., 2024), by treating the objective as a loss function, we implement a latent optimization algorithm (Algorithm 1). Second, from another perspective, the objective can serve as a verifier to score the denoising trajectories of independent samples. This enables the selection of best output, analogous to inference-time scaling in SANA 1.5 (Xie et al., 2025), for which we present a latent selection algorithm in Algorithm 2.

## 6 Experiments

### 6.1 Experimental setup

**Evaluation Metrics.** We evaluate on T2I-CompBench (Huang et al., 2025) and HRS-Bench (Bakr et al., 2023), and additionally introduce a benchmark targeting similar-object scenarios. From T2I-CompBench we use the *color*, *texture*, and *complex* tasks to assess attribute binding; from HRS-Bench we use *spatial*, *size*, and *color* for attribute binding and instance-structure formation.

Our benchmark contains two settings: *multi-class* (multiple classes, one instance per class) and *multi-instance* (multiple instances of a single class). For *multi-class accuracy* (%), a class counts as correct if the generated instance matches the target class. For *multi-instance accuracy* (%), an instance counts as correct if it matches the target class and is spatially separated from other instances. Class tags in ours benchmark prompts are drawn from COCO (Lin et al., 2014); prompt formatting appears in the Appendix. For multi-class accuracy, we enumerate all non-overlapping class pairs and report the mean. Detection uses Grounding-DINO (Liu et al., 2024) ensembled with YOLOv12 (Tian

**Algorithm 1:** Latent Optim. with ISAC

**Input:** Prompt $\mathcal{T}$, Model $\epsilon_\theta$, decoder $\mathcal{D}$, Learning rate $\eta$
**Output:** Image $I_0$ with multiple instances

1  $X_T \sim \mathcal{N}(0, I)$
2  **for** $t = T, T-1, \ldots, 1$ **do**
3     Denoise$(X_t, \mathcal{T}, \epsilon_\theta, t)$ with $\mathcal{H}^{\text{self}}, \mathcal{H}^{\text{cross}}$
4     $SA_t \leftarrow \mathcal{H}^{\text{self}}(X_t, \mathcal{T}, \epsilon_\theta, t)$
5     $CA_t \leftarrow \mathcal{H}^{\text{cross}}(X_t, \mathcal{T}, \epsilon_\theta, t)$
6     $CA_t^{\text{prop}} \leftarrow SA_t \cdot CA_t$
7     Compute $\mathcal{L}_{\text{ins}}, \mathcal{L}_{\text{cls}} \leftarrow$ Eq. 4, 6, 8, 9
8     $\mathcal{L}_t(X_t) \leftarrow$
     $\lambda_{\text{ins}}(t)\mathcal{L}_{\text{ins}}(X_t) + \lambda_{\text{cls}}(t)\mathcal{L}_{\text{cls}}(X_t)$
9     $\tilde{X}_t \leftarrow X_t - \eta \cdot \nabla_{X_t} \mathcal{L}_t(X_t)$
10    $X_{t-1} \leftarrow$ Denoise$(X_t, \mathcal{T}, \epsilon_\theta, t)$
11 $I_0 \leftarrow \mathcal{D}(X_0)$    // Decode to pixel

**Algorithm 2:** Latent Selection with ISAC

**Input:** Prompt $\mathcal{T}$, Model $\epsilon_\theta$, decoder $\mathcal{D}$, Batch size $N$
**Output:** Image $I_0$ with multiple instances

1  $X_T^{(i)} \sim \mathcal{N}(0, I), S[i] = 0, \forall i = 1, \ldots, N$
2  **for** $i = 1, \ldots, N$ **do**
3     **for** $t = T, T-1, \ldots, 1$ **do**
4       $X_{t-1}^{(i)} \leftarrow$ Denoise$(X_t, \mathcal{T}, \epsilon_\theta, t)$ with $\mathcal{H}^{\text{self}}, \mathcal{H}^{\text{cross}}$
5       $SA_t \leftarrow \mathcal{H}^{\text{self}}(X_t^{(i)}, \mathcal{T}, \epsilon_\theta, t)$
6       $CA_t \leftarrow \mathcal{H}^{\text{cross}}(X_t^{(i)}, \mathcal{T}, \epsilon_\theta, t)$
7       $CA_t^{\text{prop}} \leftarrow SA_t \cdot CA_t$
8       Compute $\mathcal{L}_{\text{ins}}, \mathcal{L}_{\text{cls}} \leftarrow$Eq. 4, 6, 8, 9
9       $\mathcal{L}_t(X_t^{(i)}) \leftarrow$
       $\lambda_{\text{ins}}(t)\mathcal{L}_{\text{ins}}(X_t^{(i)}) + \lambda_{\text{cls}}(t)\mathcal{L}_{\text{cls}}(X_t^{(i)})$
10      Score Update: $S[i] \leftarrow S[i] + \mathcal{L}_t(X_t^{(i)})$
11 $i^* = \arg\min_i S[i]$    // Best scored latent
12 $I_0 \leftarrow \mathcal{D}(X_0^{(i^*)})$      // Decode to pixel

Table 1: Quantitative comparison of ISAC (Ours) and baseline methods on HRS Benchmark, T2I-CompBench and similar-object benchmark. All tasks handle multi-class scenarios.

| Method | HRSBench | | | T2I-CompBench | | | Multi-Class Accuracy ($\uparrow$) | | | | |
|---|---|---|---|---|---|---|---|---|---|---|---|
| | Color $\uparrow$ | Spatial $\uparrow$ | Size $\uparrow$ | Color $\uparrow$ | Texture $\uparrow$ | Complex $\uparrow$ | #2 | #3 | #4 | #5 | Average |
| SD1.5 (Rombach et al., 2022) | 0.136 | 0.094 | 0.091 | 0.356 | 0.406 | 0.306 | 28% | 2% | 1% | 0% | 8% |
| + A&E (Chefer et al., 2023) | 0.149 | 0.104 | 0.101 | 0.392 | 0.447 | 0.290 | 48% | 10% | 5% | 2% | 16% |
| + SynGen (Rassin et al., 2024) | 0.159 | 0.111 | 0.107 | 0.420 | 0.479 | 0.311 | 50% | 9% | 4% | 2% | 16% |
| + InitNO (Guo et al., 2024) | 0.175 | 0.120 | 0.116 | 0.456 | 0.520 | 0.338 | 55% | 12% | 7% | 5% | 20% |
| + TEBOpt (Chen et al., 2024a) | 0.181 | 0.127 | 0.123 | 0.461 | 0.544 | 0.353 | 52% | 11% | 8% | 3% | 18% |
| + ISAC (Ours) | **0.318** | **0.263** | **0.252** | **0.683** | **0.631** | **0.354** | **65%** | **31%** | **29%** | **18%** | **36%** |
| SD3.5-M (Esser et al., 2024) | 0.425 | 0.264 | 0.209 | 0.796 | 0.726 | 0.377 | 62% | 23% | 12% | 3% | 25% |
| + A&E (Chefer et al., 2023) | 0.427 | 0.263 | 0.215 | 0.798 | 0.726 | 0.378 | 65% | 29% | 16% | 5% | 28% |
| + SynGen (Rassin et al., 2024) | 0.425 | 0.260 | 0.211 | 0.801 | 0.718 | 0.365 | 66% | 28% | 15% | 6% | 28% |
| + InitNO (Guo et al., 2024) | 0.443 | 0.275 | 0.228 | 0.810 | 0.728 | 0.378 | 77% | 31% | 17% | 7% | 33% |
| + TEBOpt (Chen et al., 2024a) | 0.438 | 0.279 | 0.220 | 0.805 | 0.730 | 0.381 | 78% | 31% | 19% | 8% | 34% |
| + ISAC (Ours) | **0.473** | **0.350** | **0.258** | **0.838** | **0.739** | **0.388** | **98%** | **51%** | **40%** | **20%** | **52%** |

et al., 2025) and YOLOE (Wang et al., 2025a) to reduce single-model errors; details of the ensemble process are in the Appendix.

Although ISAC addresses both settings, most baselines do not directly target the multi-instance task. To isolate the effect of diffusion dynamics, our main comparisons therefore focus on the multi-class setting; the multi-instance capability of ISAC is analyzed separately in Section F.

**Implementation Details.** We follow each method's official guidance (CFG and sampling steps) for SD1.5 (Rombach et al., 2022) and SD3.5-M (Esser et al., 2024). Following the baseline methods, we apply latent optimization (Algorithm 1) with ISAC. Here, the only tunable hyperparameter is the learning rate $\eta$, which is fixed to 0.01 across models. Related ablation studies are included in the Appendix. Class tag–count pairs $(\tau_i, n_i)$ are extracted from prompts using an LLM parser; in multi-class settings $n_i = 1$ for all $i$, for all prompts.

## 6.2 MAIN RESULTS

Table 1 reports results for SD1.5 and SD3.5-M backbones, which represent UNet and DiT based diffusion models respectively. Across both models, ISAC achieves the best scores on HRS-Bench (Bakr et al., 2023), T2I-CompBench (Huang et al., 2025), and on the Multi-Class Accuracy metric, consistently outperforming all training-free baselines. These gains indicate ISAC improves not only instance-structure formation (*spatial/size/multi-class*) but also its subsequent attribute binding

Table 2: Effect of objective scheduling to multi-instance generation performance. Here, we use $\lambda_{\text{cls}}(t) = 1 - \lambda_{\text{ins}}(t)$ for the purpose of balancing the two components in Eq. 10, SD1.5 (Rombach et al., 2022) is the backbone model.

| Config. | Description | $\lambda_{\text{ins}}(t)$ | $\lambda_{\text{cls}}(t)$ | Multi-Class | Multi-Instance |
|---|---|---|---|---|---|
| A | Only Instance Optimization | 1 | 0 | 10% (-26%pt) | 65% (-4 %pt) |
| B | Only Semantic Optimization | 0 | 1 | 28% (-8 %pt) | 54% (-15%pt) |
| C | Fixed Balance | 0.5 | 0.5 | 25% (-11%pt) | 60% (-9 %pt) |
| D | Semantic-to-Instance, Hard | $H(t - T/2)$ | $1 - H(t - T/2)$ | 19% (-17%pt) | 52% (-17 %pt) |
| E | Semantic-to-Instance, Soft | $1 - t/T$ | $t/T$ | 21% (-15 %pt) | 55% (-14%pt) |
| F | Instance-to-Semantic, Hard | $1 - H(t - T/2)$ | $H(t - T/2)$ | 35% (-1 %pt) | 67% (-2 %pt) |
| G | Instance-to-Semantic, Soft (Ours) | $t/T$ | $1 - t/T$ | **36%** | **69%** |

Table 3: Alternative similarity metrics for the proposed MPO in Eq. 7. We use soft transition schedule for all configurations.

| Loss type | Multi-Class | Multi-Instance |
|---|---|---|
| MAE | 9 % (-27%pt) | 55% (-14%pt) |
| KL | 16% (-20%pt) | 60% (-9%pt) |
| IoU | 20% (-16%pt) | 61% (-8%pt) |
| MPO (Ours) | **36%** | **69%** |

(*color/texture/complex*). Additional quantitative results in Table 13 shows consistent and model-agnostic gains with ISAC latent optimization algorithm.

Qualitative results in Figure 5 shows that in most cases SD1.5 (Rombach et al., 2022), A&E (Chefer et al., 2023), InitNO (Guo et al., 2024) (1st, 3rd rows) and SynGen (Rassin et al., 2024) (1st, 2nd row) fail to generate all 3 instances. Even succeed in generating 3 instances, they often wrongly allocate the class labels—A&E (Chefer et al., 2023) and InitNO (Guo et al., 2024) (2nd row) and SynGen (Rassin et al., 2024) (3rd row)—leading to another instance missing problem. Also in the case of SD1.5 (Rombach et al., 2022), A&E (Chefer et al., 2023) and InitNO (Guo et al., 2024) (2nd row), the left dog with brown hair indicates semantic features from adjacent instances flooded to it, which is a common failure mode of instance merging. In contrast, ISAC successfully generates 3 decoupled instances with distinct appearances, demonstrating its effectiveness in multi-instance generation.

| Baseline | + Attend-and-Excite | + SynGen | + InitNO | + ISAC (Ours) |
|---|---|---|---|---|

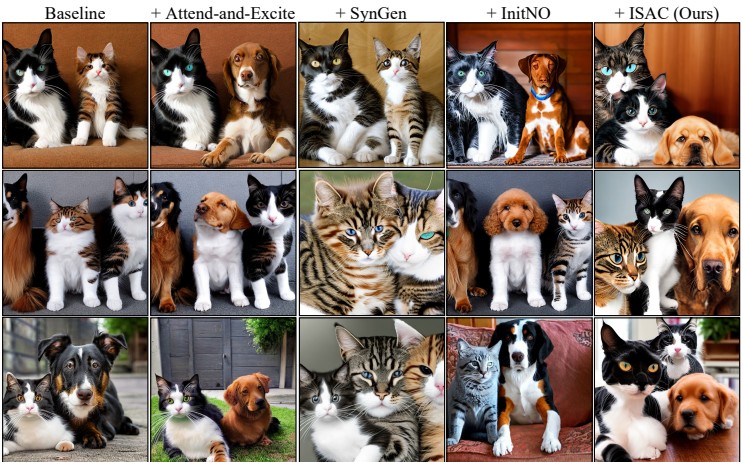

Figure 5: Qualitative comparison using SD1.5 (Rombach et al., 2022) as a backbone and added attention control methods. For all cases,"A photo of two cats and a dog" is an input prompt.

## 6.3 Ablation Study

We ablate the fundamental effect of our method design to instance structure formation and instance-semantic binding. Since our similar-object benchmark handles that basic scenarios and as it supports instance count-wise report, we majorly use it for ablation study and discussions.

**Contribution of Loss Components.** We ablate the instance-decoupling loss $\mathcal{L}_{\text{ins}}$ and instance-semantic binding loss $\mathcal{L}_{\text{cls}}$ by selectively disabling each term (Table. 2). With SD1.5, optimizing *only* $\mathcal{L}_{\text{ins}}$ (Config. A) yields 10% multi-class and 65% multi-instance accuracy; optimizing *only* $\mathcal{L}_{\text{cls}}$ (Config. B) gives 28% and 54%; using constant weights (Config. C) gives 25% and 60%. Both components are therefore necessary for balanced performance.

**Scheduling instance-to-class dynamics.** We compare weighting schedules for $(\lambda_{\text{ins}}(t), \lambda_{\text{cls}}(t))$ in Table. 3. The *soft two-phase* schedule (Ours; Config. F) attains **36%** multi-class and **69%** multi-instance accuracy, outperforming a *reversed* class-to-instance schedule (Config. D), which underperforms even the constant baseline (Config. C). This supports aligning optimization with diffusion dynamics: prioritize instance separation early, then refine semantic assignment. Also, *soft*

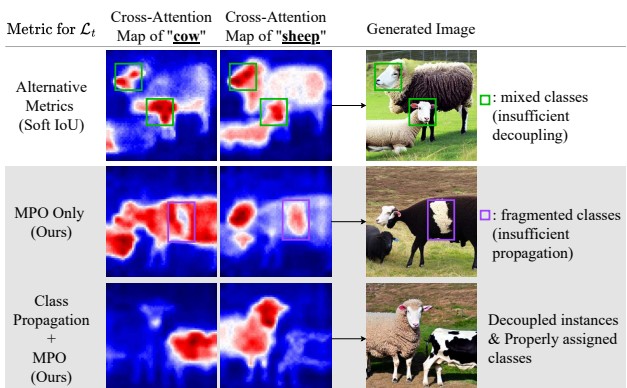

Generation Prompt: "A photo of a **sheep** and a **cow**"

Figure 6: Effect of MPO and Class Propagation. Compared to the baseline metric, our full method produces better instance decoupling and class consistency in multi-class prompts.

*two-phase* marginally outperforms the *hard* transition (Config. E), verifying the soft two-phase design is valid for temporal dynamics of diffusion models.

**Effectiveness of Maximum Pixel-wise Overlap (MPO) and Class Propagation.** Replacing MPO with alternative similarities substantially degrades accuracy (Table 3). With MPO, multi-class and multi-instance accuracies reach **36%** and **69%**, respectively, versus **20%/61%** for IoU, **16%/60%** for KL, and **9%/55%** for MAE. These results indicate that MPO's peak-overlap penalty enforces strict local exclusivity, improving both instance separation and semantic binding.

Figure 6 further illustrates these effects. Optimizing with global metrics (e.g., Soft IoU) yields overlapping cross-attention for different classes ("cow"/"sheep"), producing mixed activations and semantic leakage in the image. Using MPO alone removes most overlaps but can leave fragmented, under-propagated class activations. Combining MPO with class propagation resolves both issues, yielding well-localized attention maps and cleanly separated classes in the final image.

## 7 CONCLUSION

We presented ISAC, a training-free objective that explicitly separates instance formation from semantic binding, aligning inference control with the dynamics of diffusion models. By enforcing early structural exclusivity and subsequent semantic alignment, ISAC overcomes the common failure modes of instance merging and omission. Across diverse benchmarks and backbones, ISAC consistently improves multi-instance fidelity and integrates seamlessly with layout-guided or fine-tuned pipelines. These results suggest that instance-first dynamics are a fundamental principle for multi-object generation, opening new directions for extending control to video, medical imaging, and distilled diffusion models.

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
