# OpenReview forum: "ISAC: Training-Free Instance-to-Semantic Attention Control for Improving Multi-Instance Generation"
_ICLR.cc/2026/Conference — ICLR 2026 Conference Withdrawn Submission_

### Official Review · Reviewer_WcDh · 2025-10-27

**Soundness:** 3
**Presentation:** 3
**Contribution:** 3
**Rating:** 4
**Confidence:** 4

**Summary:**

This paper proposes ISAC (Instance-to-Semantic Attention Control) to address the failure of text-to-image (T2I) diffusion models in generating multiple instances in complex scenes. By analyzing the temporal dynamics of diffusion models, the authors observe that semantic signals are underdeveloped before instance structures form. Based on this, they design a two-phase control objective: Phase 1 forms instance structures using structural signals, and Phase 2 binds semantics to these structures. The method is evaluated on multiple benchmarks (T2I-CompBench, HRS-Bench) and a custom similar-object scenario, demonstrating strong performance. Overall, the work is technically sound, innovative, and well-supported by experiments.

**Strengths:**

1. Methodological novelty: ISAC introduces a dynamic-aware two-phase objective that decouples instance structure formation and semantic binding, effectively addressing missing or merged instances in multi-instance generation.

2. Technical completeness: The method includes global foreground mask creation, K-means clustering for instance structure, Maximum Pixel-wise Overlap (MPO) for strict separation, and semantic binding, making it a comprehensive and practical approach. It can be applied as a loss function or verifier and is compatible with latent optimization and latent selection.

3. Extensive experiments: Evaluation on multiple benchmarks and scenarios, with metrics covering both instance separation and attribute binding, as well as qualitative analysis, supports the effectiveness of ISAC.

**Weaknesses:**

1. The description of the method in the paper needs to be clearer.

2. The examples presented in the paper are relatively limited; they are mostly cats and dogs, with only a few examples of other categories (SD1.5) shown in the appendix.

3. This training-free approach generally tends to degrade image quality, and the method lacks evaluation of image quality.

4. The method introduces significant time and memory overhead, increasing inference memory from 23GB to 75GB and substantially prolonging inference time.

5. Some formatting issues exist in the paper, such as the page number on page 6 being enclosed within a citation box.

**Questions:**

1. Could the authors provide more intuitive explanations or visual illustrations for some of the formulas in the method section?

2. Could a relatively complex example be included? For instance, if one wants to generate a person wearing many accessories and equipment, can such overlapping cases be generated correctly?

3. Most of the examples shown in the paper are cats and dogs. Could the authors include examples from other domains? Additionally, it would be helpful to show more results using stronger generators, such as SD3.

4. Does the proposed method affect the quality of the generated images? Could metrics such as FID or aesthetic scores be provided to evaluate potential quality degradation?

5. The method introduces substantial memory overhead. Compared to directly using existing L2I models or RL-finetuned models, is this overhead justified? Can the method achieve significantly better results?

6. There are some formatting issues in the paper, such as the page number on page 6 being enclosed within a citation box.

---

> ### Author Response · Authors · 2025-11-20
>
> We are grateful for your positive evaluation, especially your recognition of ISAC’s 'methodological novelty' and 'technical completeness.' Although we are withdrawing the submission, we would like to briefly share how we have expanded our method clarity, evaluation scope, and efficiency analysis in response to your constructive questions.
>
> **1. Method Clarity and Intuitive Illustrations.** We agree that the method description can be made more accessible. In our revision, we are adding step-by-step visualizations of attention maps, similar to the dynamics analysis in Figure 4 and Figure 14. These illustrations explicitly show how ISAC ensures instance layouts emerge early (Phase 1) and how semantics are bound to those specific instances later (Phase 2), clarifying the formulas in the method section.
>
> **2. Complex Examples and Diversity (Beyond Cats/Dogs).** You raised a valid point regarding the prevalence of animal examples. To address your request for complex human-centric generation, we have included qualitative results for "person" (e.g., "two boys") in Figure 13 of the Appendix. Also, we have expanded our qualitative evaluation to include diverse domains such as food, sports equipment, and vehicles (e.g., "broccoli, hot dog, pizza", "skis, baseball glove") as shown in Figure 5 and Figure 15.
>
> **3. Image Quality (FID and Aesthetic Scores).** We appreciate your query regarding potential quality degradation. We will measure FID and CMMD on baselines with/without ISAC.
>
> **4. Overhead and Justification vs. L2I/Fine-tuned Models.** ISAC is designed to complement, not just compete with, these models. As shown in Tables 10 and 11, adding ISAC to GLIGEN (L2I) or IterComp (RL-finetuned) consistently boosts their performance. It resolves semantic bleeding and overlapping failures that even these specialized models struggle with. For limited resources, we propose the latent selection variant (Algorithm 2). As shown in Table 11, this approach offers substantial accuracy improvements (e.g., 88% avg. multi-instance accuracy on Flux) with negligible memory overhead compared to the optimization method.

---

### Official Review · Reviewer_BQsb · 2025-10-30

**Soundness:** 2
**Presentation:** 1
**Contribution:** 2
**Rating:** 4
**Confidence:** 5

**Summary:**

The paper proposes a training-free instance-to-semantic attention control for improving multi-instance generation. It identifies that instance features are formed first followed by the semantic features which are then bound to the instances. Based on this it proposes a new MPO loss for separating instances and class propagation loss for binding attributes to classes. Experiments are conducted on T2I-Compbench and HRS-Bench for multi-class and multi-instance setups.

**Strengths:**

The proposed method works across UNet and Diffusion Transformer models and can be combined with layout-guided methods.

It introduces a new Maximum pixel-wise Overlap criterion to enforce mutually exclusive boundaries between instances based on the observation that instance structures form early in the diffusion timesteps.

**Weaknesses:**

The evaluation of the proposed method is insufficient contrary to the title and claims made in the paper. It needs to be evaluated with other similar instance generation methods that can handle more complex tasks such as MIGC and InstanceDiffusion since the proposed method can be combined with those.


The multi-instance evaluation setup is weak as it only uses instances of the same class and does not evaluate multiple instances of two classes such as “A photo of two dogs and three cats”

In L210, it mentions that the instances should be mutually exclusive, how does this method work for small objects. All qualitative results are shown only for objects that cover a significant portion of the image.

Quantitative evaluation of the proposed method is only performed on simple object classes like animals and not on persons or indoor objects. Section C.3 mentions that these are omitted due to the difficult nature of the objects and so the effectiveness of the method is limited beyond a certain set of classes. The qualitative results in Appendix Figure 13 for different objects needs to be compared with prior works as they seem to be cherry picked.


The proposed latent optimization method is slow, sensitive to the noise seeds and more importantly does not work robustly for all cases, sharing the inherent limitations of prior inference-based latent optimization techniques.


The presentation of the paper is not clear and it can be improved further. The results for Algorithm 2 are not discussed in the paper and so it can be moved to Appendix or removed. Table 13 in the Appendix including the latency and memory costs should be moved to the main paper.

Minor:
Typo in L10 of Algorithm 1 and L1200

**Questions:**

See weaknesses above

---

> ### Author Response · Authors · 2025-11-20
>
> We appreciate your rigorous review and constructive feedback. Your comments on evaluation breadth and stability were particularly helpful in identifying areas where our claims required stronger empirical backing. Below, we outline how we are addressing your concerns regarding comparisons, stability, and evaluation scope in our ongoing revision.
>
> **1. [W1] Comparisons to Stronger Controllers (e.g., MIGC, InstanceDiffusion).** We agree that comparing with complex task handlers is valuable. However, distinct from MIGC or InstanceDiffusion which require training or auxiliary modules, ISAC is a training-free method designed to complement them. We have demonstrated this synergy on fine-tuned models (Table 10; Appendix) and layout controllers (Table 11; Appendix). For instance, adding ISAC to GLIGEN consistently improves counting, color, and spatial metrics on HRS-Bench, proving its effectiveness as a plug-and-play module for stronger baselines.
>
> **2. [W2, W3, W4, W5] Evaluation Protocol and Object Diversity.** You raised valid concerns about the single-class focus of our multi-instance metric and the limited object classes in our quantitative results. To address mixed-class scenarios (e.g., "two dogs and three cats"), we refer to our results on the T2I-CompBench Numeracy task (Table 9), where ISAC achieves strong performance on complex, multi-category prompts. Regarding the omission of "person" or "indoor objects" in our benchmark (Section C.3), this was a decision to ensure reliable automatic detection for evaluation, not a limitation of ISAC itself. As shown in our qualitative results (Figure 13), ISAC successfully generates people ("two boys") and indoor scenes ("cushion on couch"), reinforcing structural cues regardless of object type. We will expand our main paper evaluation to explicitly include these categories.
>
> **3. [W5] Stability and Seed Sensitivity.** We respectfully clarify the concern that latent optimization induces instability. Our data indicates that ISAC actually stabilizes generation. In our stability analysis using 30 random seeds on T2I-CompBench (Table 15), ISAC reduced the standard deviation by ~3–4× (SD1.5: 46.5\% $\pm$ 5.1\%, SD1.5 w/ ISAC: 55.0\% $\pm$ 1.4\%) while significantly improving mean accuracy.
>
> **4. [W6] Readability.** We fully accept your suggestions to improve readability. In the next version, we will move Algorithm 2 (Latent Selection) to the appendix to streamline the methodology section and bring the efficiency analysis (Table 13) into the main paper to make the cost-performance trade-off transparent.

---

### Official Review · Reviewer_EAL4 · 2025-10-30

**Soundness:** 3
**Presentation:** 3
**Contribution:** 2
**Rating:** 6
**Confidence:** 4

**Summary:**

This paper proposes the ISAC framework, which employs a two-stage objective function: the first stage leverages self-attention to achieve instance-structure separation, and the second stage utilizes cross-attention to accomplish semantic binding. The approach incorporates the MPO metric to optimize spatial exclusivity and semantic correspondence in multi-instance generation. Experiments conducted on multiple mainstream text-to-image models demonstrate that the framework significantly improves the quality of multi-instance generation.

**Strengths:**

1. Based on the observation that in the denoising process of diffusion models, spatial instance structures emerge before clear semantics materialize, the proposed stage-wise separation algorithm is more reasonable and better aligned with this generative process compared with prior work.
2. The effectiveness of the method is validated across multiple popular text-to-image models (e.g., SD1, SD2, SD3, SDXL, PixArt), showing consistent improvements in multi-instance generation quality.

**Weaknesses:**

1. Compared with the original approach, the proposed method may introduce additional computational overhead. The inclusion of latent optimization and VLM models requires larger VRAM and increases inference time.
2. The work lacks comparative experiments with LLM + Layout methods.

**Questions:**

1. Potential Reduction in Layout Diversity. Given the conclusion stated in Line 210 that “A total of N instances should occupy mutually exclusive, N distinct regions”, I am concerned that such a constraint may limit layout diversity. For example, if the input prompt describes “a cup and three ice cubes”, the proposed method may tend to generate images where the ice cubes appear outside the cup, rather than depicting them inside the cup. This could potentially reduce the natural compositional variety expected in certain prompts. Could the authors clarify whether the framework allows flexibility for compositional layouts where overlap or containment is required?
2. Overlapping Semantic Regions. For prompts such as “a cat standing behind a transparent glass”, a particular image region may need to visually represent both the cat and the glass simultaneously. In such cases, the assumption that each instance should occupy a distinct, non-overlapping region could hinder accurate rendering, especially when transparency is involved. How does the proposed method address these complex spatial relationships while preserving semantic accuracy?
3. Baseline Selection and Efficiency Comparison. Since the proposed framework leverages an LLM to identify each instance within a textual description, it seems natural to compare against a baseline where an LLM is used to parse scene layouts and is subsequently combined with layout-to-image generation methods. In addition, direct efficiency comparisons (e.g., VRAM usage, inference time) with these layout-based approaches would make the evaluation more comprehensive.

---

> ### Author Response · Authors · 2025-11-20
>
> Thank you for your insightful assessment, particularly for highlighting how our stage-wise separation aligns well with the generative process of diffusion models. In light of our decision to withdraw and refine the paper, we would like to briefly clarify the points you raised regarding computational overhead (W1+Q3) and layout constraints (W2+Q1), as these have driven significant updates in our work.
>
> **1. Computational Overhead (Latent Optimization vs. Selection).** We acknowledge your concern regarding the additional computational cost introduced by latent optimization. As ISAC is a loss objective, its optimization cost is inherent to the inference-time optimization paradigm, comparable to existing methods (Table 1). To address this, our revision highlights latent selection (Algorithm 2) as an alternative application of ISAC. This approach utilizes ISAC as a verifier to select the best trajectory without gradient updates, significantly reducing the overhead (VRAM) while maintaining improved generation quality.
>
> **2. Comparison with LLM + Layout Methods.** You raised a valid point regarding the lack of comparison with "LLM + Layout" baselines. We have verified that ISAC is highly complementary to these methods. As shown in our Appendix (Table 11), applying ISAC on top of GLIGEN (a representative layout-to-image model) consistently improves performance across counting, color, spatial, and size metrics. This demonstrates that ISAC can effectively refine the coarse bounding boxes provided by layout models into dense instance masks, further enhancing the fidelity of layout-guided generation. We will explicitly highlight this synergy in the revised manuscript.
>
> **3. Limitations on Layout Diversity and Transparency.** We appreciate your thoughtful questions regarding "ice cubes in a cup" or "transparent glass," which challenge the assumption of mutually exclusive regions. We acknowledge this as a limitation. ISAC uses the proposed MPO metric to enforce strict separation, which effectively prevents the common failure mode of objects merging. However, as you correctly noted, this strict exclusivity can hinder the rendering of transparent objects or complex containments where pixel-level semantic overlap is physically required (z-axis ambiguity). We will add a discussion on this trade-off to our “Limitations” section.

---

### Official Review · Reviewer_iFMs · 2025-11-02

**Soundness:** 3
**Presentation:** 3
**Contribution:** 3
**Rating:** 6
**Confidence:** 3

**Summary:**

This work presents Instance-to-Semantic Attention Control (ISAC), a framework designed to enhance multi-instance generation in text-to-image diffusion models. It addresses a key limitation of current models—their difficulty in handling scenes with multiple distinct instances. ISAC introduces a hierarchical, training-free approach that separates instance formation from semantic binding during the denoising process. By tackling the challenges of instance separation and semantic alignment, the framework significantly improves the accuracy and fidelity of generated images in complex, multi-object scenarios.

**Strengths:**

The proposed approach effectively addresses common failures in multi-object generation by decoupling instance formation from semantic binding, enabling more accurate and coherent image synthesis in complex scenes.

The proposed method is model-agnostic and can be complementary to fine-tuned models and also enhance existing layout-guided models.

The paper is well-structured, and the supplementary material contributes meaningfully to both understanding and reproducibility of the proposed approach.

**Weaknesses:**

While the paper introduces some interesting ideas, its contribution is somewhat limited by the fact that object count and layout consistency in multi-object image generation are already being actively explored in the literature. Given the growing number of existing approaches, the novelty of the proposed method appears incremental, and broader comparisons with related works could further clarify its distinct advantages.

The conclusion section is relatively underdeveloped; it lacks a thorough articulation of the significance and impact of the contribution. Additionally, the vision for future work is only addressed in the supplementary material, which limits its visibility and integration into the main narrative of the paper.

**Questions:**

Please, see the previous comments.

Several existing works explicitly address the challenges of object counting and coherent layout generation in multi-object image synthesis. Notable examples—some of which are already cited in the paper—include CountGen, COUNTLOOP, and IMAGHarmony. These contributions reflect an active research landscape in this area. Extending the discussion to consider such or similar approaches would help better situate the proposed method within the broader context and clarify its unique contributions.

CountGen. Make It Count: Text-to-Image Generation with an Accurate Number of Objects (CVPR2025)
COUNTLOOP. Iterative Agent Guided High Instance Image Generation
IMAGHarmony. Controllable Image Editing with Consistent Object Quantity and Layout

---

> ### Author Response · Authors · 2025-11-20
>
> We would like to express our gratitude for your thoughtful review and for explicitly recognizing the effectiveness of our decoupled instance-to-semantic approach. Although we have decided to withdraw the paper to further refine the manuscript, we wish to share how your valuable feedback regarding novelty (W1+Q1) and presentation (W2) is shaping our revision.
>
> **1. Novelty: Instance-First Principle vs. Existing Methods.**
> We wish to clarify that ISAC is distinct from count verifiers or layout controllers; it establishes an instance-first control principle aligned with diffusion dynamics.
>
> - ISAC vs. Count-Supervised Methods: Approaches like CountGen rely on fine-tuned modules or external models that are effective only after semantic features emerge. By contrast, ISAC intervenes early to stabilize instance structure before semantics solidify. Our experiments show ISAC outperforms these methods even given the same count cues. We will expand Table 8 (Appendix) to include the additional baselines you suggested.
> - ISAC vs. Layout Controllers: While layout controllers partition semantics, they often struggle with overlapping boxes. ISAC is model-agnostic and complements these controllers by refining coarse boxes into dense instance masks. As shown in Table 11 (Appendix), applying ISAC to GLIGEN improves performance across counting, color, spatial, and size metrics.
>
> **2. Improving Conclusion and Future Work.**
> We fully accept your feedback regarding the manuscript's structure. In the next version, we are rewriting the conclusion to better articulate the significance and limitations of our method.

---

### Note · Authors · 2025-11-20

**Comment:**

We sincerely appreciate the time and effort the Area Chair and all reviewers (iFMs, EAL4, BQsb, WcDh) have dedicated to evaluating our work.

**Withdrawal Confirmation:**

I have read and agree with the venue's withdrawal policy on behalf of myself and my co-authors.